# Microbial Hotspots in Lithic Microhabitats Inferred from DNA Fractionation and Metagenomics in the Atacama Desert

**DOI:** 10.3390/microorganisms9051038

**Published:** 2021-05-12

**Authors:** Dirk Schulze-Makuch, Daniel Lipus, Felix L. Arens, Mickael Baqué, Till L. V. Bornemann, Jean-Pierre de Vera, Markus Flury, Jan Frösler, Jacob Heinz, Yunha Hwang, Samuel P. Kounaves, Kai Mangelsdorf, Rainer U. Meckenstock, Mark Pannekens, Alexander J. Probst, Johan S. Sáenz, Janosch Schirmack, Michael Schloter, Philippe Schmitt-Kopplin, Beate Schneider, Jenny Uhl, Gisle Vestergaard, Bernardita Valenzuela, Pedro Zamorano, Dirk Wagner

**Affiliations:** 1Center for Astronomy and Astrophysics, Technische Universität Berlin, 10623 Berlin, Germany; f.arens@tu-berlin.de (F.L.A.); heinz@tu-berlin.de (J.H.); yunhahwang@alumni.stanford.edu (Y.H.); j.schirmack@tu-berlin.de (J.S.); 2GFZ German Research Centre for Geosciences, Section Geomicrobiology, Telegrafenberg, 14473 Potsdam, Germany; dlipus@gfz-potsdam.de (D.L.); beate.schneider@bam.de (B.S.); 3Leibniz-Institute of Freshwater Ecology and Inland Fisheries (IGB), Department of Experimental Limnology, 16775 Stechlin, Germany; 4School of the Environment, Washington State University, Pullman, WA 99163, USA; 5German Aerospace Center (DLR), Institute of Planetary Research, 12489 Berlin, Germany; Mickael.Baque@dlr.de; 6Environmental Microbiology and Biotechnology, Department of Chemistry, University of Duisburg-Essen, 45141 Essen, Germany; till.bornemann@uni-due.de (T.L.V.B.); jan.froesler@uni-due.de (J.F.); rainer.meckenstock@uni-due.de (R.U.M.); mark.pannekens@uni-due.de (M.P.); alexander.probst@uni-due.de (A.J.P.); 7German Aerospace Center (DLR), Microgravity User Support Center (MUSC), 51147 Cologne, Germany; jean-pierre.devera@dlr.de; 8Department of Crop and Soil Science, Washington State University, Pullman, WA 99164, USA; flury@wsu.edu; 9Department of Crop and Soil Science, Washington State University, Puyallup, WA 98371, USA; 10Department of Chemistry, Tufts University, Boston, MA 02155, USA; samuel.kounaves@tufts.edu; 11Department of Earth Science & Engineering, Imperial College London, London SW7 2AZ, UK; 12GFZ German Research Centre for Geosciences, Section Organic Geochemistry, 14473 Potsdam, Germany; kai-mangelsdorf@gfz-potsdam.de; 13Research Unit for Comparative Microbiome Analysis, Helmholtz Zentrum München, Ingolstädter Landstr. 1, 85764 Neuherberg, Germany; johan.saenz@helmholtz-muenchen.de (J.S.S.); schloter@helmholtz-muenchen.de (M.S.); 14Research Unit Analytical BioGeoChemistry, Helmholtz Zentrum München, Ingolstädter Landstr. 1, 85764 Neuherberg, Germany; schmitt-kopplin@helmholtz-muenchen.de (P.-S.K.); jenny.uhl@helmholtz-muenchen.de (J.U.); 15Federal Institute for Materials Research and Testing (BAM), 12205 Berlin, Germany; 16Department of Health Technology, Technical University of Denmark, 2800 Lyngby, Denmark; gisle.vestergaard@helmholtz-muenchen.de; 17Laboratorio de Microorganismos Extremófilos, Instituto Antofagasta, Universidad de Antofagasta, Av. Angamos 601, Antofagasta 1240000, Chile; extremelifehunters@gmail.com (B.V.); pedro.zamorano@uantof.cl (P.Z.); 18Institute of Geosciences, University of Potsdam, Karl-Liebknecht-Str. 24-25, 14476 Potsdam, Germany

**Keywords:** hyperarid, habitat, desert ecology, extremophile, endolith, hypolith

## Abstract

The existence of microbial activity hotspots in temperate regions of Earth is driven by soil heterogeneities, especially the temporal and spatial availability of nutrients. Here we investigate whether microbial activity hotspots also exist in lithic microhabitats in one of the most arid regions of the world, the Atacama Desert in Chile. While previous studies evaluated the total DNA fraction to elucidate the microbial communities, we here for the first time use a DNA separation approach on lithic microhabitats, together with metagenomics and other analysis methods (i.e., ATP, PLFA, and metabolite analysis) to specifically gain insights on the living and potentially active microbial community. Our results show that hypolith colonized rocks are microbial hotspots in the desert environment. In contrast, our data do not support such a conclusion for gypsum crust and salt rock environments, because only limited microbial activity could be observed. The hypolith community is dominated by phototrophs, mostly Cyanobacteria and Chloroflexi, at both study sites. The gypsum crusts are dominated by methylotrophs and heterotrophic phototrophs, mostly Chloroflexi, and the salt rocks (halite nodules) by phototrophic and halotolerant endoliths, mostly Cyanobacteria and Archaea. The major environmental constraints in the organic-poor arid and hyperarid Atacama Desert are water availability and UV irradiation, allowing phototrophs and other extremophiles to play a key role in desert ecology.

## 1. Introduction

For terrestrial ecosystems, the spatial heterogeneity of abiotic and biotic properties has been well described on the micro- [1], meso-, and macroscale [2,3]. Consequently, microbial activities in soils are not uniform but highly scattered and mainly follow the availability of nutrients. Thus the rhizosphere, detritusphere, drilosphere, or aggregate surfaces have been described as hotspots for microbial activity, whereas bulk soil has been considered as a cold spot with mostly dormant and inactive microbes as a result of low nutrient availability [4].

Desert ecosystems are different from terrestrial environments in temperate regions, where hotspots are largely dependent on plant vegetation and the availability of nutrients, mostly organic carbon [1]. Desert ecosystems such as the Atacama are characterized by an extreme lack of water and organic matter, high UV irradiation, and high salt concentrations, and thus they are an extreme challenge to any type of life [5,6,7,8,9]. Our hypothesis is that these factors inhibit the development of microbial activity hotspots in desert environments more than the availability of nutrients. Further, it has been recognized that biologically receptive mineral substrates may play a significant role on where microbial colonization and activity can be found [5,10,11,12,13,14,15,16,17,18]. Thus, we postulated that in desert environments zones underneath or within rocks, particularly beneath quartz rocks, and within gypsum crusts and halite nodules represent hotspots for microbial activity, because they form refuges for microorganisms in response to inhospitable environmental conditions. These hotspots are characterized by hypolithic and endolithic microorganisms, as here microbiota are protected from major stressors occurring in such desert ecosystems. Therefore, this study focuses on localities where other researchers have observed the existence of islands of habitability [19,20,21,22,23,24,25]. However, the question remains whether these islands are only a location where improved habitable conditions exist or represent actual hotspots of microbial activity. In these desert regions, microbial life generally exists only at levels near the detection limit [10,26].

Observations and results from previous investigators have significantly advanced our current understanding of the composition and behavior of microbial populations in the Atacama Desert [19,20,21,22,23,24,25]. Biological soil crusts, for example, have been identified as a hotspot of microbial activity and a main driver of nutrient cycling in the less arid regions of the Atacama Desert [27]. Cyanobacteria seem to be prevalent in both the arid and hyperarid regions and are known to both colonize quartz rocks using a hypolithic lifestyle (e.g., [28,29]) and, via halite nodules, using an endolithic lifestyle (e.g., [15,30]). There are fewer studies that investigated gypsum crusts, but it is now clear that at least some of them are inhabited as well (e.g., [31,32,33,34,35]). These studies, which are based on total DNA analysis, show Proteobacteria, Acidobacteria, Bacteroidetes, Chloroflexi, and Firmicutes as dominating microbial phyla. None of these studies identified the living and potentially active part of the microbial communities. Therefore, this raises the question if lithic habitats in the hyperarid regions of the Atacama Desert take the role of hotspots similar to the way a biological soil crust does in the less arid regions of the Atacama. At this point, only limited data are available on the distribution and abundance of active microorganisms among the microbial populations in the Atacama Desert, which is critical to identify microbial hotspots, and thereby minimizes our ability to make accurate predictions on the survivability and viability of specific communities. To fill this gap, expand the current state of knowledge on Atacama microbial populations, and specifically obtain data helping us to discern between active and non-active microbial communities, or even between modern and past communities, we employed a multifaceted approach combining a novel DNA extraction method, different biomarker measurements, and the calculation of metagenomic replication rates.

To not only assess the overall microbial community, but specifically identify active microbiota, we used a novel method by Alawi et al. [36] that allowed to distinguish between intracellular DNA (iDNA) and extracellular DNA (eDNA). The iDNA pool derives from intact cells, while the eDNA pool is comprised of DNA not contained within a cellular membrane—thus not active. This method was already successfully applied in different environments, including the Atacama and Namib deserts [10,26,37], showing significant differences in the microbial community structure between the two DNA pools. We used an amplicon-based sequencing approach to assess the diversity of bacteria and archaea. Further, we performed genome-resolved metagenomic analyses to infer whether microbial species were replicating and containing viral signals at the time of sampling, both indications of activity. Results from the DNA and metagenomic analyses were then correlated to the results obtained from our ATP analyses (where we were also able to distinguish between internal and external ATP), and phospholipid fatty acid (PLFA) and metabolite analyses to gain further insights on the activity of the microbial community and to determine whether these preselected localities are indeed microbial hotspots with elevated microbial activity compared to the surrounding sediments.

## 2. Methods

### 2.1. Study Sites and Sampling Procedure

Samples were collected during a 2015 and 2017 sampling campaign along a moisture gradient from the coast into the Atacama Desert at specific locations deemed to represent different microhabitats. The Costal Soil site with the hypoliths sample (CS-HL) is located 21 km south of Antofagasta and about 1 km inland from the shoreline (481 m.a.s.l., S 23.8247, W 70.4888) representing the least arid location. The Alluvial Fan site with hypolith and gypsum crust (AL-HL, AL-GG) samples is located on the eastern slope of the coastal range, about 22 km inland (847 m.a.s.l., S 24.0008, W 70.2958). The Yungay Salar with the halite nodules (YS-HN) is located about 60 km inland in the hyperarid core of the Atacama Desert (964 m.a.s.l., S 24.08679, W 69.91321). Mini-PAM measurements (pulse amplitude modulated, Heinz WALZ GmbH, Effeltrich, Germany) have been used to detect the presence of chlorophyll and photosynthetic activity [38,39,40]. PAM fluorometry uses light flashes to infer the presence of photosynthetic pigments and determine the photosynthetic efficiency of photosynthetic system II (PSII), with chlorophyll a at its center. Chlorophyll is therefore used as an indicator for the presence of photosynthesizing organisms, and the gypsum crust indeed indicated the presence of some phototrophic organisms, which were thus included in the analyses. In addition, sediment samples were collected from selected nearby locations to establish a background for the putative microbial hotspots.

### 2.2. Sample Preparation

Samples were abraded with a diamond grinding head mounted on an electric hand drill. For every sample, a new grinding head was used which was sterilized beforehand with 100% alcohol and flamed over a Bunsen burner. The workplace and every tool were carefully sterilized with 70% alcohol. The powder was collected on aluminum foil and transferred in an autoclaved Eppendorf tube. Sediment samples used here as background samples were collected and prepared for analyses as described in Schulze-Makuch et al. [26].

### 2.3. Mineralogy

Mineral analysis was conducted on oven-dried (60 °C for 24 h), homogenized, and powdered samples. Therefore, we used a powder X-ray diffractometer (Bruker Desktop D2, Massachusetts, USA) at the Laboratory of Mineralogy of the Technische Universität Berlin. The X-ray source was Cu Kα radiation (K-alpha1 = 1.540598 Å, K-alpha2 = 1.54439 Å) with a performance of 30 kV and 10 mA. A scanning range from 5° to 90° 2Θ and a step interval of 0.013° 2Θ with a step-counting time of 1 s were applied. The mineral constituents were identified with the software High Score (PANalytics, Irvine, CA, USA) and the mineral database PDF-2 (International Centre of Diffraction Data, Newtown Square, PA, USA), which allowed for a semi-quantitative evaluation. It also determined the relative intensity ratios of the identified minerals. 

### 2.4. Moisture Sorption Isotherms and Water Activity

Oven-dried samples (105 °C for 24 h) were used to determine the moisture sorption isotherms. After cooling in a desiccator, samples were placed into the sample chamber of an AquaSorb Isotherm Generator (Decagon Devices, Inc., Pullman, WA, USA). A flow of water-saturated air was passed over the samples, causing the samples to continuously wet up. Air flow was set to 250 mL/min, and operating temperature was 25 °C. Water content and water activity were measured simultaneously with a high precision magnetic force balance and a chilled-mirror dew point sensor, respectively. The maximum water activity was set to a_w_ = 0.88–0.89. After this limit was reached, samples were dried by passing dry air through the sample chamber. Again, water content and water activity were measured simultaneously while the sample dried. These measurements yield a drying and wetting isotherm of moisture sorption (Figure 1). Measurements were duplicated. Standard solutions of KCL, NaCl, and LiCl were used to verify water activity measurements.

The moisture sorption isotherms were used to estimate the water activity of the samples when they were taken in the field. The measured gravimetric water contents of the field samples were used to read their estimated water activity from the moisture sorption isotherms. The moisture sorption isotherms did not show hysteresis, so that a single water activity was obtained for each water content. The water activity of the hypolith was a_w_ = 0.725 and that of the halite was a_w_ = 0.753, indicating that the halite was water-saturated at the time of sampling.

### 2.5. DNA Isolation and Sequencing

#### 2.5.1. Extraction of Extracellular (e) and Intracellular (i) DNA

The eDNA and iDNA from abraded (AL-HL, YS-HN) scraped-off (CS-HL) and ground material (AL-GG) were extracted according to the method from Alawi et al. (2014), which was slightly modified and described in detail recently [10,26]. Both iDNA and eDNA were separated from the sample matrix, in this cases sediment and rock powder, using agitation. Briefly, 6 g of sample material was mixed with 0.6 g PVPP and 7 mL sodium phosphate (NaPNa_2_PO_4_) buffer in a sterile 50 mL falcon tube. Samples were then placed on ice for 1 min, followed by agitation on a horizontal shaker for 5 min at 150× rpm. The samples were cooled on ice for 3 min, and the shaking procedure was repeated. The resulting slurry was centrifuged for 10 min at 500× *g*. The supernatant, containing the eDNA fraction, was collected and stored on ice, followed by the addition of another 3.5 mL of sodium phosphate buffer and another round of agitation on the orbital shaker and centrifugation. The procedure was repeated two more times, with the pellet re-suspended in decreasing amounts (2 and 1.5 mL) of sodium phosphate buffer. At the conclusion, the pellet contained the iDNA released from the intact cells, while the eDNA remained in the collected supernatant. The supernatant was pooled and divided into three technical replicates and centrifuged for 1 h at 4643× *g*.

Then, iDNA was extracted from the resulting cell pellet using the Power Soil^®^DNA isolation kit (Mo Bio Laboratories Inc., Carlsbad, CA, USA), according to manufacturer’s recommendations. The supernatant containing the eDNA was filtered through a 0.2 μm syringe filter to remove any remaining cell components or other contaminants. The eDNA was then collected by adding silica particles (ρ: 1.3 g mL^−1^) together with guanidine hydrochloride (GuaHCl, 6M, 4× volume of extract) and was shaken for 45 min at 175 × rpm followed by incubation on ice for 10 min and centrifugation for 10 min at 4643× *g*. The supernatant was discarded, and the silica pellet was washed with 70% ethanol buffer. Finally, the eDNA was eluted from the silica matrix using TE buffer.

Separate DNA extraction methods were applied to be able to discern between active microbial communities (iDNA) and free DNA (eDNA) that may originate from the lysis of dead cells and thus represents an indicator for previous colonization. Alternatively, eDNA may occur in the environment due to autolysis, active secretion systems of living cells, or horizontal gene transfer.

#### 2.5.2. Quantitative PCR Analysis (qPCR) and Amplicon Sequencing

qPCR was performed using a CFX Connect Real-Time PCR detection system (Bio-Rad, Hercules, CA, USA) in duplicates of iDNA and eDNA and corresponding blank controls using KAPA SYBR^®^FAST qPCR Master Mix (KAPA BIOSYSTEMS, Wilmington, MA, USA). DNA was amplified with the universal primers 331F and 797R [41] and the following cycling parameters: initial denaturation at 95 °C, 3 min, followed by 40 cycles (95 °C, 3 s; 60 °C, 20 s; 72 °C, 30 s; 80 °C, 3 s). The correlation coefficient for the standard curves was ≥0.99, and the PCR efficiency was on average 98%. The standard was a known concentration of a 16S rRNA gene PCR fragment of *Bacillus subtilis.*

#### 2.5.3. 16S rRNA Gene Amplicon Pool Preparation for Illumina MiSeq Sequencing

PCR amplification targeted the hypervariable region V4 of the 16S rDNA using forward primer, 515F: 5′-GTGCCAGCMGCCGCGGTAA-3′ and reverse primer 806R: 5′-GGACTACHVGGGTWTCTAAT-3′, each of them specified with 6-bp tags. PCR amplification was performed in at least triplicate in 25 µL reactions (2.5 µL 10 × PCR buffer, 0.5 µL ultrapure dNTP-mix (5 mM), 0.25 µL of each primer (10 mM), 1.5 µL MgCl_2_ (25 mM), 2–5 µL template, 0.25 µL HotStar Taq polymerase (Qiagen, Hilden, Germany) under the following conditions: initial denaturation at 95 °C for 15 min, followed by 10 cycles of 95 °C for 30 s, 65 °C at −1 °C/cycle for 30 s, 72 °C for 45 s, 25 to 40 cycles (depending on DNA concentration of the different samples) of 95 °C for 30 s, 56 °C for 30 s, 72 °C for 45 s, and a final extension step of 10 min at 72 °C. The reactions were pooled, purified with Agencourt AMPure XP magnetic beads (Beckman Coulter, Brea, CA, USA), and quantified with a Qubit Fluorometer (Invitrogen, Thermo Fisher Scientific, USA). Purified PCR amplicons from all samples were pooled in equimolar ratios to a final concentration of approximately 120 ng µL^−1^. The amplicon library was sequenced using the Illumina MiSeq v. 3 kit (2 × 300 bp) at Eurofins Scientific (Constance, Germany). Sequencing generated between 241,188 and 819,564 reads, with an average number of 515,884 reads per sample (Appendix A). Sequencing reads from this study were deposited in the European Nucleotide Archive under the project accession number PRJEB43972.

The microbial community composition was analyzed by high-throughput sequencing of the extracellular and intracellular DNA pools (e- and iDNA) independently, which resulted in 8,254,152 sequencing reads in the 16 samples in total, including replicates. Processing of 16S rRNA gene MiSeq data through filtering, merging, and chimera and contamination removal trimmed results to 4,729,436 reads in the final dataset. The eDNA pool encompassed 45% of the total number of sequences, while the iDNA pool encompassed 55% of the total number of sequences. The number of read counts was between 69,772 and 598,392 reads, with an average of 295,589 reads per sample (more details are given in Appendix A). Libraries were demultiplexed using Cutadapt [42] while having the requirements that primer sequences had a maximum error rate of 10%, and a barcode was required to have a phred score higher than Q25 without any mismatches. Sample sequences were further processed using the DADA2 pipeline [43]. Reads were truncated (250—forward, 200—reverse) and quality-filtered before the error model was generated. Reads were required to have a minimum length of 200 bp after dereplication, sample inference, and merging of the paired-end reads. The orientation of the reads was standardized by calculating the hamming distances of the sequences and their reverse complement. The sequence table was constructed, and chimeras were removed using a de novo approach. The amplicon sequence variants were assigned to the SILVA taxonomy (v132) [44] using vsearch [45] as utilized in the framework of QIIME2 [46].

#### 2.5.4. Diversity and Statistical Analyses

A sequence cut off number of 40 was set based on artifact sequences occurring in the positive control (*E. coli* DNA) sample. ASVs (amplicon sequence variants) with 40 or less sequences (<0.01% across all samples) were removed from the ASV table. The ASV table was subsampled to a depth of 69,772 sequences for subsequent alpha diversity analyses, which represents the sequencing count for the lowest number of sequences in an individual sample. Community richness (observed number of ASVs) and evenness (Shannon diversity index) were calculated using the RTK and Vegan packages in R [47,48]. Differences in alpha diversity measures (number of ASVs and Shannon diversity index) between pools and across habitats were evaluated using analysis of variance (ANOVA) and Kruskal–Wallis test in Past 3.17 software. No statistical analysis could be conducted for AL-GG samples due to a low number of replicates (*n* = 1).

The ASVs shared between samples or unique for a specific sample set were identified using an online tool and visualized utilizing Venn and Euler diagrams (http://www.venndiagrams.net/#tab-6807-3; accessed on 15 October 2020). For beta diversity analysis, the un-rarefied ASV table was normalized using Hellinger transformations. Bray–Curtis distances between samples were calculated using the PAST 3.17 software [49]. Samples were visualized using nonmetric dimensional scaling (NMDS). Heatmaps showing relative abundances were constructed using the Heatmap2 tool and devtools library in R [50]. Dendrogram trees accompanying heatmaps were generated using hierarchical clustering based on Bray–Curtis distances utilizing the Heatmap2 package [51].

Indicator species for each habitat were determined by comparing occurrence and abundance of each ASV across all samples. ASVs occurring in all habitats at a relative abundance of >0.1% were considered core ASVs, while ASVs with a relative abundance of >1% in one of the habitats and a relative abundance of <0.1% in the remaining habitats were considered habitat specific ASVs. These values were selected based on previous analyses [52].

#### 2.5.5. Read-Based Metagenomics, Genome-Resolved Metagenomics, and In Situ Replication Rates

A total of 0.5 g of material was scraped from the hypolith samples using sterile scalpels. Particles were introduced in Lysing Matrix E tubes (MP Biomedicals, Eschwege, Germany), and a bead beating and phenol-chloroform-isoamylalcohol (PCI) based protocol was used to extract the metagenomic DNA [53]. DNA from two extractions were pooled for each library preparation. The 500 bp DNA fragments were generated using a E220 focused-ultrasonicator (Covaris^®^ Inc., Woburn, MA, USA). Illumina libraries were prepared using a NEBNext Ultra DNA library prep kit and NEBNext Multiplex Oligos. DNA fragments with an insert size between 500 and 700 bp were selected and purified based on Agencourt^®^ AMPure^®^ XP (Beckman-Coulter, Danvers, MA, USA) instructions. DNA size and concentration were measured with the Fragment AnalyzerTM (Advanced Analytical, Ankeny, IA, USA) and the PicoGreen method. Negative extraction controls had DNA concentrations below the detection limit (0.5 ng/μL). Libraries were pooled equimolarly, and 15 pM was spiked with 1% PhiX (Illumina, San Diego, CA, USA). Libraries were sequenced as paired-end (2 × 300 bp) using an Illumina MiSeq system. Negative extraction controls treated in an identical manner yielded 100-fold fewer reads and were taxonomically distinct from the soil samples [26].

Genome resolved-metagenomics was successfully performed for samples from the Coastal Soil hypoliths. In brief, MiSeq reads were quality filtered using BBduk version 37.09 (https://sourceforge.net/projects/bbmap/, accessed on 1 October 2017) and sickle v 1.33 (https://github.com/najoshi/sickle, accessed on 1 October 2017) and assembled using metaSPADES [54]. Genes were predicted for scaffolds ≥1 kb and annotated against UniRef100 [55]. The taxonomy of annotated genes was used to determine a consensus taxonomy for the scaffolds. Coverage of scaffolds was calculated using Bowtie2 mapping in sensitive mode [56]. Genomes were binned using both only tetranucleotide frequency-based (abawaca, version 1.07 (github.com/CK7/abawaca, accessed on 16 April 2018) and ESOM [57] as well as a differential coverage utilizing a binning algorithm (maxbin2, [58]) and aggregated using DAS_Tool [59]. Binned genomes were manually curated using taxonomy, GC content, and coverage of scaffolds prior to scaffolding error correction using ra2 [60] and an additional manual curation iteration. Genomes were dereplicated using dRep with default parameters [61]. Completeness and contamination of final bins were determined using CheckM [62], and bins were considered high-quality for completeness >90% and contamination <10%. In situ genome replication rates (iRep) were calculated for high quality bins allowing a maximum of three mismatches per mapped read [61]. Rank abundance curves were calculated based on genes annotated as ribosomal protein S3 (*rpS*3) against UniRef100 [55] using DIAMOND [63]. Abundances were calculated for the entire scaffold carrying the rpS3 gene as described above. Functional and metabolic capacities of the genomic bins were determined using PhenDB [64] as well as hidden Markov models for key enzymes involved in chemolithoautotrophic pathways as used in Anantharaman et al. [65].

Viral signals from the metagenomes were predicted using VirSorter (Roux et al. 2015) using default settings and -diamond flag. Extraction of repeats and Cas genes were performed using CRISPRCasFinder [66] (https://crisprcas.i2bc.paris-saclay.fr/CrisprCasFinder/Index, accessed on 15 January 2019); only repeats with evidence level 4 or higher were selected for spacer extraction. CRISPR spacers from high quality genomes were extracted using MetaCRAST [67] with flags -d 3 -l 60 -c 0.9 -a 0.9 -r and matched against the viral signal by BLASTing [68] and filtering for 0.8 similarity (similarity = alignmentLength* Identity)/QueryLength). The protein sequences in viral contigs were extracted using Prodigal [69] and identified by BLASTing them against UniRef100 9).

### 2.6. ATP Analyses

Sediment samples were placed in a sterile autoclave bag and crushed to smaller pieces (up to a maximum diameter of approximately 1 cm) using a hammer. Six grams of sediment or crushed rock samples were introduced into a 50 mL centrifuge tube, and 5 mL of ice-cold sodium phosphate buffer (0.12 M Na_2_HPO_4_, NaH_2_PO_4_, pH = 8.0) were added. Samples were shaken on an orbital shaker for 5 min at 150 rpm, cooled on ice for 3 min, and shaken again for another 5 min. Samples were then centrifuged at 4 °C and 500 g for 10 min. The supernatants containing the dislodged cells as well as extracellular ATP (eATP) were recovered in a 15 mL centrifuge tube, and 1 mL of sodium phosphate buffer was added to the sediment samples. The procedure was repeated 3 times, and supernatants were collected. The collected suspensions were centrifuged at 4 °C and 4600 × *g* for 60 min. Supernatants containing the eATP fraction were recovered in 15 mL centrifuge tubes. Cell pellets containing intracellular ATP (iATP) were re-suspended in 1–4 mL of sodium phosphate buffer, and the particles in the solution were allowed to settle for approximately 30 min before samples were subjected to ATP analysis. All samples were processed in triplicates. ATP was quantified using the luciferase-based BacTiter-Glo^TM^ microbial cell viability assay (Promega). Measurements were carried out in opaque 1.5 mL microcentrifuge tubes according to the manufacturer’s protocol, using a 6-point calibration curve with ATP concentrations ranging from 10 pM to 1 μM. The sample solutions containing iATP or eATP, respectively, were measured undiluted. A 0.12 M sodium phosphate buffer was used as a blank and for diluting the standard series. Then 100 μL of sample solution, blank, or standard were mixed with 100 μL of BacTiter-Glo^TM^ reagent, which was prepared on the day before measurement and kept at room temperature until measurements were performed. Five minutes after mixing, luminescence was recorded using a Glomax 20/20 luminometer (Promega, Madison, WI, USA). All assays were analyzed in duplicate. ATP concentrations were normalized to sample weight.

### 2.7. Phospholipids/Phospholipid Fatty Acids Analysis (PLFA)

#### 2.7.1. Extraction and Column Separation

Approximately 60 g of freeze-dried ground samples were extracted by using a flow blending system with a 200 mL mixture of methanol/dichloromethane/ammonium acetate buffer (2:1:0.8, pH 7.6) modified after Bligh and Dyer [70,71]. The solvent extract was transferred into a separation funnel for phase separation. As internal standards, 50 µg of 1-myristyl-(D27)-2-hydroxy-sn-glycerol-3-phosphocholine was added. For phase separation, dichloromethane and water were added to achieve a ratio of 1:1:0.9 of methanol/dichloromethane/ammonium acetate buffer mixture, and the organic phase was removed. The water phase was re-extracted twice with dichloromethane, and all organic phases were combined. Subsequently, the obtained sediment extract was separated into four fractions of different polarity (low polar lipids, free fatty acids, glycolipids, and phospholipids (PLs)). Two columns were used in sequence. The upper column was filled with 1 g silica gel (63–200 µm) and topped with 0.5 g of sodium sulfate. The lower column was filled with 1 g Florisil. According to the method described by Zink and Mangelsdorf [71], the low polar fraction was eluted with 20 mL of chloroform, the free FAs with 50 mL of methyl formiate blended with 12.5 µL of glacial acetic acid, and the glycolipid fraction with 20 mL of acetone. After removal of the Florisil column, the PLs were eluted with 25 mL of methanol from the silica column. To improve the recovery of PLs, the silica column was rinsed with 25 mL of a methanol/water mixture (60:40), and the extract was captured in a separation funnel. Then 15 mL of dichloromethane and 3.5 mL of water were added for phase separation (methanol/dichloromethane/water, 1:1:0.9), the organic phase was removed, and the water phase was re-extracted twice with dichloromethane. Finally, the organic phases were combined, and all fractions were evaporated to dryness and stored at −20 °C until analysis.

#### 2.7.2. Detection of Phospholipid Fatty Acids (PLFA)

After rock powder extraction and column separation, the PL fraction was used for PLFA analysis following an ester cleavage procedure outlined by Mueller et al. [72]. Subsequently, the resulting PLFAs were measured by gas chromatography–mass spectrometry (GC–MS). The GC–MS measurements were conducted on a Trace GC Ultra (Thermo Electron Corporation) coupled to a DSQ Thermo Finnigan Quadrupole MS (Thermo Electron Corporation). The GC was equipped with a cold injection system operating in the splitless mode and a SGE BPX 5 fused-silica capillary column (50 m length, 0.22 mm ID, 0.25 µm film thickness) using the following temperature conditions: initial temperature 50 °C (1 min isothermal), heating rate 3 °C/min to 310 °C, held isothermally for 30 min. Helium was used as carrier gas with a constant flow of 1 mL/min. The injector temperature was programmed from 50 to 300 °C at a rate of 10 °C/s. The MS operated in the electron impact mode at 70 eV. Full-scan mass spectra were recorded from *m*/*z* 50 to 650 at a scan rate of 1.5 scans/s. Cell biomass in cells gSed^−1^ was calculated using the conversion factor from PLFAs to cells (20,000 cells/pmol PLFA) after Balkwill et al. [73].

### 2.8. Metabolites

#### 2.8.1. Soil and Rock Extraction

A detailed description of the sediment sample extraction is given in Schulze-Makuch et al. [26]. Hypoliths from the Coastal Soil location, gypsum crusts from the Alluvial Fan, and halite nodules from Yungay Salar were first cracked and rinsed with 300 µL of methanol (Chromasolv LC-MS grade methanol, Sigma Aldrich, Taufkirchen, Germany), before they were crushed in 300 µL methanol and ultrasonicated for 3 min. The afterwards transferred supernatants were evaporated and the samples resolved in acidified (pH 2) purified water (MilliQ-Integral, Merck KGaA, Darmstadt, Germany) and hydrochloric acid (32%, p.a., Merck KGaA, Darmstadt, Germany). Afterwards, each sample was extracted using C18 OMIX Tips (100 µL, Agilent Technologies, Waldbronn, Germany) following a standardized protocol of conditioning, washing, and eluting pipette cycles with a resulting sample volume of 500 µL. The eluates were kept at −20 °C for further analysis.

#### 2.8.2. ESI(−) FT-ICR-MS Analysis

Mass spectra were acquired in negative ionization mode using a SolariX Qe FT-ICR-MS equipped with a 12 T superconducting magnet and coupled to an Apollo II electrospray ionization source (Bruker Daltonik, Bremen, Germany). Methanolic rock extracts were continuously infused with a flow rate of 120 µL h^−1^, whereas soil samples were diluted 1:20 with methanol beforehand. Spectra accumulated 300 scans within a mass range of 147 to 1200 *m*/*z*. An internal calibration was performed with a mass accuracy of <0.1 ppm, and peaks with a signal to noise ratio >4 were picked. Formula assignment was performed with in-house written software (NetCalc) using a network approach to calculate chemical compositions containing carbon, hydrogen, and oxygen, as well as nitrogen and/or sulfur [74]. The mass accuracy window for the formula assignment was set to ±0.5 ppm, and the assigned formulas were validated by setting sensible chemical constraints (N rule; O/C ratio ≥1; H/C ratio ≤ 2n + 2 (maximum possible carbon saturation, with n defined as C*n*H*n*+2 for any formula), double bond equivalents) in conjunction with isotope pattern comparison. Results were visualized by the use of van Krevelen diagrams in which the hydrogen to carbon ratio (H/C) was plotted against the oxygen to carbon ratio (O/C). The different bubble sizes represent the intensity of the characteristic molecular formula within the respective sample.

## 3. Results

### 3.1. Field Scanning and Mineralogical Results of Sampling Sites

We collected lithic samples from four field sites, which we selected based on visible evidence of microbial colonization. When questionable, we also used field scanning with a PAM (pulse amplitude modulated) instrument, which detects the presence of photosynthetic pigments and chlorophyll via photometry. Two hypolithic habitats (CS-HL and AL-HL) were identified, one site with gypsum crusts (AL-GG), and at the most hyperarid site, the Yungay Salar, we found halite nodules colonized by endoliths (YS-HN) (Figure 2). The hypolithic substrate at the Alluvial Fan (AL) location was composed of 100% quartz, while the hypolithic substrate at the Coastal Soil (CS) location consisted of 84% quartz and 16% feldspar (Figure 3a). A sufficient sample volume could not be retrieved from AL-HL to conduct all analyses for that field site. The gypsum crusts at AL consisted of 93% gypsum and 7% quartz (Figure 3b), while the salt nodules at the Yungay Salar (YS) were composed of 95% halite, 3% gypsum, and 2% quartz (Figure 3c,d). Measured water content was low in both the CS quartz rocks and YS halite nodules (0.1% and 0.7% by weight, respectively), but relatively high in the gypsum crusts at AL (14.9% by weight; the high water content of the gypsum crusts derives from the crystal water contained in gypsum, which was already lost below the drying temperature of 105 °C).

In our analysis, we also compared those sites to background sedimentary sites in the same vicinity of the sample locations (Figure 2). The sediment samples were dominated by alkali feldspar and plagioclase with minor amounts of quartz, chlorite, and amphibole, and their water content was less than 1% by weight for the CS location and less than 0.4% by weight for the other sediment sampling locations (AL, RS, and YU) [26].

### 3.2. Composition of the Microbial Community Based on e- and iDNA Analyses

In this study the bacterial and archaeal abundance, diversity, and community composition across four Atacama habitats were evaluated by analyzing the extracellular and intracellular DNA (e- and iDNA) pools using 16S rRNA amplicon sequencing and quantitative PCR (qPCR). This approach allowed us to obtain an estimate of the overall prokaryotic distribution to reconstruct a detailed description of the living microbial community and to gather insights on past microbial populations.

Quantitative PCR demonstrated that bacterial and archaeal abundance (as 16S rRNA gene copy numbers per gram of material), based on e- and iDNA pools, were significantly (ANOVA, *p* < 0.05) higher in the two quartz rock samples containing the hypoliths (10^8^–10^10^ gene copies) compared to gypsum crusts (10^3^ gene copies), the halite-dominated salt rocks (10^3^–10^4^ gene copies), and the unconsolidated sediments (10^1^–10^4^ gene copies). The 16S rRNA gene copy numbers of eDNA and iDNA were found to have non-significant differences according to a *t*-test (*p* > 0.05, Table 1).

Microbial diversity varied within each habitat and between iDNA and eDNA pools, as indicated by Shannon indices (Table 1). In eDNA pools from quartz hypoliths, the Shannon indices were generally higher (4.27 ± 0.86 SD and 3.51 ± 0.08 SD) but not significantly different (ANOVA: *p* = 0.07, Kruskal–Wallis test for equal median: *p* = 0.05) compared to eDNA pools from halite nodules (2.91 ± 0.68 for eDNA) and gypsum crusts (3.02). Shannon indices for iDNA pools were similar across hypolith and halite nodule habitats (AL-HL = 2.89 ± 0.53, CS-HL = 3.09 ± 0.83, YS-HN = 2.43 ± 0.39) and highest in the gypsum crust sample (3.45). No statistically significant differences were observed. The richness in the iDNA pool, numbers of observed amplicon sequence variants (ASVs), rarified to a sequencing depth of 69,772 across all samples, were overall higher in the two hypolith environments than in the gypsum crusts and halite nodule samples (Appendix A, Figure 4). Highest richness (ASV numbers) was observed in the CS-HL sample (651 observed ASVs for eDNA and 599 observed ASVs for iDNA). The iDNA pools of AL-GG (207 observed ASVs) and YS-HN (288 observed ASVs) had the lowest ASV counts. Analysis of variance (ANOVA) revealed statistically significant differences in ASV numbers between iDNA pools in CS-HL, and Al-HL and YS-HN habitats (both *p* < 0.05). Differences in diversity between pools were assessed using t-tests. Shannon indices and ASV numbers were similar between iDNA and eDNA pools (all *p* > 0.05) across three of the four habitats. In the costal soil hypolith habitat (CS-HL), Shannon indices between the two pools varied slightly (4.27–3.09); however, the difference was found to be non-significant (*p* = 0.09).

Our analysis revealed distinct prokaryotic community structures in the quartz rocks containing hypoliths (CS-HL, AL-HL), the gypsum crusts (AL-GG), and the halite nodules (YS-HN), showing habitat-specific communities. In addition, we observed major differences in taxonomic profiling between eDNA and iDNA pools for the AL-GG and YS-HN samples, as only 12% and 11% of ASVs were shared among the two pools (Figure 4a). In the hypoliths a greater amount of ASVs (55% for CS-HL and 38% for AL-HL) was shared among the pools (Figure 4a).

The total microhabitat microbial communities, comprised of eDNA and iDNA pools, were dominated by bacteria (98.5–99.9%), with the one exception being the Yungay halite nodules, where a large fraction of archaea (about 55.5%) was discovered. The eukaryotic fraction of the microbial community was not evaluated, because based on the PLFA analyses, only traces of the respective markers were found in the gypsum crusts and halite nodules, and those that could indicate a fungal presence were interpreted as of cyanobacterial origin (see PLFA section below).

While the overall hypolith communities were particularly enriched in Cyanobacteria of the order Nostocales, gypsum crust and halite nodule populations were found to be more diverse. Evaluation of the Yungay Salar halite nodule community revealed high abundances of halophilic archaea, Cyanobacteria, and several taxa of Proteobacteria, including Pseudomonadales and Betaproteobacteriales (Figure 4b). Proteobacteria were almost exclusively identified in the eDNA pool and thus represent an indicator for a previously existing community in this location. The gypsum crusts AL-GG samples were dominated by Proteobacteria, Actinobacteria, and Bacteroidetes. Similar, to the Yungay samples, Proteobacteria were especially abundant in the eDNA and were comprised of the orders Methylophilales, Rhizobiales, Caulobacterales, and Pseudomonadales, suggesting a methylotrophic community may have existed at some point (Figure 4b).

Phylogenetic classification and ordination analysis using nonmetric multidimensional scaling based on Bray–Curtis dissimilarities (NMDS) further supported these observations, as samples from the two hypolith habitats formed distinct clusters, with small differences between the DNA pools, while YS-HN and AL-GG samples were more scattered and clustered pool specific (Appendix A).

### 3.3. Active Microbial Community Based on iDNA Pool Evaluation

This study explicitly aimed to explore and further characterize living and potentially active microbial life. Accordingly, the presence, abundance, and distribution of intact cells, independent of their physiological state, here represented by the intracellular DNA (iDNA) pool, was analyzed. To further evaluate how much the active communities in the surveyed habitats resemble each other or differ from each other, we specifically looked at taxonomic ASVs occurring across all four habitats (core ASVs) and ASVs only occurring in one of the four habitats (habitat specific ASVs). ASVs only occurring in specific habitats can be considered specialists for this type of environment [75]. Using specific criteria (described in detail in the Methods section) we were able to identify five ASVs that occurred in all four evaluated habitats. The majority (four) of these ASVs belonged to Proteobacteria and could be classified as Burkholderiaceae (ASV 5), *Methylobacterium* (ASV 9), Methylophilaceae (ASV 4), and *Pseudomonas* (ASV 12) (Figure 5). The remaining shared ASVs (20) could be classified as *Nocardia* (Actinobacteria). With the exception of ASV 4, ASV 5, and ASV 9, which were enriched in the AL-GG habitat (up to 18%), these core ASVs were only detected at low frequencies across the assessed samples (Figure 5).

Detailed characterization of the iDNA community highlighted potential habitat specific ASVs and provided novel insights into which microbial processes microorganisms may actively contribute in the investigated habitats of the Atacama Desert. Cyanobacteria of the order Nostocales were especially abundant (64% iDNA reads in CS-HL and 61% iDNA reads in AL-HL) in the two hypolith environments (Figure 5 and Figure 6). A closer analysis indicated that the hypolith communities diverged on the ASV level, as the most abundant ASV in the CS-HL habitat could be classified as *Aliterella* (46%), while the AL-HL community was dominated by Chroococcidiopsaceae (51.7%) and uncultured Nostocales ASVs (2.6%–4.9%) (Figure 5 and Figure 6). In addition, the AL-HL community was defined by a greater occurrence and abundance of Chloroflexi ASVs (up to 2.6%), while the Actinobacteria ASVs *Rubrobacter* (2.7%) and *Conexibacter* (1.8%) were more enriched in the CS-HL iDNA pool (Figure 6 and Appendix A). A distinct series of ASVs linked to cyanobacterial habitat specialists could be identified in each of the hypolith habitats, suggesting unique cyanobacterial communities may exist at each location. Two *Truepera* ASVs were exclusively identified in the CS-HL iDNA pool, designating this radiation-tolerant taxon as a potential indicator species for Coastal Soil hypoliths. In contrast to the hypolith environments, Cyanobacteria were less abundant in the salt rock (YS-HN) and gypsum crust (AL-GG) iDNA pools, but were each characterized by a unique microbial composition. The YS-HN iDNA pool was dominated by archaeal Halobacteriales (44%) and bacterial Nostocales (32%), which are both related to phototrophic microorganisms, pointing towards a halo-tolerant and phototrophically active community. Inter-habitat comparison demonstrated the enriched Nostocales ASV to be different from those discovered in the hypoliths (Figure 5 and Figure 6). Unfortunately, the limited phylogenetic resolution of the 16S rRNA amplicon did not allow classification of this ASV below the order level. The halophilic archaea were affiliated with various *Halococcus* (24%), *Halomarina* (2%), unclassified Halomicrobiaceae (5%), and unclassified Halobacteriales (10%) ASVs and could only be identified in this particular environment. A third group of microorganisms were only detected (13% relative abundance) in the YS-HN samples and clustered close to the highly halophilic *Salinibacter* (order Bacteroidetes). Besides the frequently detected methylotrophic taxa Methylophilaceae and the unclassified Burkholderiaceae (as described above), the Alluvial Fan gypsum crust iDNA community was especially defined by the presence of various heterotrophic phototrophs, mostly Actinobacteria and Chloroflexi ASVs, that were found to be unique to this habitat (Figure 5). *Crossiella*, *Conexibact*er, *Rubrobacter*, and uncultured Acidimicrobiia each represented more than 2.5% of the overall population (Figure 4 and Appendix A), while *Blastococcus* and uncultured Kallotenuales (up to 10%; Figure 5 and Appendix A) were more abundant. The most dominant identified specialist ASV was *Segetibacter* (13%). Finally, the AL-GG iDNA pool was the only sample to contain reads affiliated with the hyperthermophilic taxon *Thermobaculum*.

Microbial communities in the unconsolidated surface (0–5 cm) sediments (Coastal Soil, Alluvial Fan, Red Sands, and Yungay Salar) have previously been described in Schulze-Makuch et al. [26]. Overall, sediment communities at the Coastal Sand and Alluvial Fan sites were found to differ from those observed in the hypoliths, as not Cyanobacteria, but halophilic archaea belonging to the Halomicrobaceae were most abundant in the CS surface sediment’s iDNA pool, and Bacillaceae were particularly enriched in the AL surface sediment´s iDNA pool.

Only limited cyanobacterial signatures could be detected across the unconsolidated sediment communities, as the highest relative abundance was observed at the Yungay site. While Cyanobacteria were also enriched in the iDNA fraction of the Yungay salt nodules, the communities differed at a lower taxonomic level (Class Sericytochromatia vs. Nostocales), and the salt nodules were also characterized by a high relative abundance of halophilic archaea, which were not present in the unconsolidated sediments.

Nevertheless, there are some similarities in community composition between sediment and habitat specific samples. One such case was the frequent identification of Burkholderiaceae in both the Alluvial Fan hypoliths and the unconsolidated sediment communities as well as the occurrence of Chloroflexi belonging to the Kallotenuales in the iDNA pools representing the Alluvial Fan hypoliths, gypsum crust, and the unconsolidated sediment communities.

### 3.4. Genome-Resolved Metagenomics

Genome-resolved metagenomic analyses were done on the collected samples to further elucidate active microbial populations. However, only the hypolith samples yielded enough DNA for a successful analysis. Four library duplicate extractions of the two hypolithic samples from CS-HL yielded on average approx. 1.7 Gbp of sequences per library. Through genome-resolved metagenomics, we retrieved eight and nine draft genomes from the two hypolithic samples, respectively. After dereplication, three cyanobacterial draft genomes and one *Rubrobacter* draft genome from CS-hypolith sample 1 and four cyanobacterial draft genomes from CS-hypolith sample 2 remained (Figure 7, Appendix A). The completeness of the population genomes estimated using checkM [62] ranged between 82.11% and 99.41%, and the contamination ranged between 0.71% and 3.84%. Detailed statistics on the population genomes can be found in Appendix A.

To infer replication measures of the bacterial high quality draft genomes, we calculated the index of replication (iRep) [61]. This method uses the coverage slope between origin of replication and the terminus region to assess the average number of replication forks occurring in situ for a given genome. The iRep values ranged between 1.45 and 2.03 (Figure 7), providing evidence for actively replicating bacterial populations, assuming one origin of replication in each bacterial genome. An iRep value of 1.5, for instance, might indicate that on average, half of the cells of a specific population (i.e., at a species level) is replicating. Interestingly, the known radiation-tolerant Actinobacteria bin *Rubrobacter* [76] was predicted to have the highest replication measure of genomes, with every genome having one replication fork on average. All other recovered high-quality genomes were classified as Cyanobacteria and all showed active replication of the genomes. Consistent with previous studies of hypoliths in deserts [77,78,79], the two communities were dominated by Cyanobacteria, with *Chroococcidiopsis* being the dominant genus. *Chroococcidiopsis* has been previously characterized to be tolerant to desiccation and ionizing radiation [79,80,81]. Since Cyanobacteria usually retrieve their cellular carbon from carbon fixation, we conclude that active genome replication indicates metabolic activity and active carbon fixation by these organisms. Other detected microbes (based on the presence and coverage of ribosomal protein S3) in lower abundance were Actinobacteria, Deinococcus-Thermus, and Chloroflexi. Although previous studies have analyzed the metagenome of the hypoliths [77,82], here we present the first genome-resolved metagenomic study of the hypolithic communities that characterizes the functional diversity coupled with computational evidence of in situ replication.

The functional capacities of the high quality bins were determined using PhenDB [64], a machine learning-based software that predicts microbial phenotypes using models that are trained on the NCBI RefSeq genomes database. Key pathways that were present in all *Chroococcidiopsis* bins included photoautotrophy, phosphonate degradation, phosphorus recycling, and urea degradation. Consistent with previous metagenomic studies of hyperarid desert microbial communities [83,84], no complete nitrogen fixation pathways were present in any of the bins, although published *Chroococcidiopsis* genomes (i.e., *Chroococcidiopsis thermalis* CP003597) found in other environments have nitrogen fixation pathways present, and *Chroococcidiopsis* has been thought to be a nitrogen fixing Cyanobacteria [85]. One metagenome assembled genome (MAG) (Hypolith 1_3, Appendix A) had a complete assimilatory nitrate reduction pathway, suggesting that nitrate could be the possible main source of cellular nitrogen.

Both metagenomes from the hypolith samples contained viral (phage and prophage) signals, which were predicted using VirSorter [86]. Of the 31 predicted phage contigs in hypolith sample 1, three were predicted to be prophages according to VirSorter in contrast to two out of 44 for the hypolith sample 2. CheckV [87] determined 4%, 5.3%, and 56% of the predicted signal to be “high quality”, “medium quality”, and “low quality”, respectively. The quality of the remaining 34.7% of the predicted viral contigs could not be determined. CheckV predicted one and six signals to be of prophages from hypolith sample 1 and 2, respectively. Five out of the eight retrieved genomes contained extensive CRISPR array systems with the Cas genes present. Notably, upon matching spacers with virus sequences from both hypolith samples, two putative viral sequences (one from each sample) matched spacers from two distinct cyanobacterial host populations, one per sample (Appendix A). If these viruses are indeed only present in one hypolith sample (it is possible that less abundant viruses fail to assemble and be detected), the evidence that a virus from one site interacts with hosts from two separate sites suggest that these cyanobacterial populations are relatively slowly evolving. In addition, 9 and 53 protein sequences were found in virus 1 and virus 2, respectively. The quality of virus 1 could not be determined using checkV due to the lack of viral genes detected, while virus 2 was of “medium quality” with completeness between 64 and 100% and a contamination of 3.83%. Of these, 8 and 31 were either uncharacterized or hypothetical proteins, respectively. Identified proteins from virus 2 included transposases, lysozymes, transcriptional regulators, chromosome partitioning protein, and Clp proteases. No putative auxiliary metabolic genes were identified.

### 3.5. ATP

ATP analyses were used as an additional indicator as to whether the present microbial communities are active. Our methodology allowed the separation of intracellular ATP (iATP), indicative of viable cells, and extracellular ATP (eATP), indicative of ATP remaining in the soil after cell lysis. iATP levels were about five orders of magnitude higher in the hypolith at CS-HL compared to the other rock samples and surrounding sediment samples, which were all at about the same level of 10^−14^ mol/g (Figure 8). The eATP levels were about three orders of magnitude higher in the hypoliths compared to the other samples. Based on ATP analyses, only the hypoliths appear to be an active habitat.

### 3.6. PLFA

Bacterial membrane phospholipid esters (PLs) and their side chain fatty acids (PLFAs), obtained after ester cleavage, are mostly used as life markers for living bacteria, since they are only stable in living bacteria over longer periods of time [88,89]. PLFAs were found in all microhabitat samples, suggesting the presence of bacterial life in the collected lithic rock samples. The bacterial cell biomass calculated from recovered PLFA ranged from 6.2 × 10^6^ for the hypoliths and 1.3 × 10^6^ for the gypsum nodules to 8.5 × 10^5^ cells gSed^−1^ for the salt rocks. The highest diversity of PLFAs was found in CS-HL with nearly 60 different PLFAs from 6 different fatty acid groups. These variations were in the same range as for the surrounding sediment samples (Figure 9). In contrast, the diversity of PLFAs encountered in the gypsum crust (AL-GG, 45 PLFAs from 5 groups) and salt rocks (YS-HN, 28 PLFAs from 4 groups) was lower than at CS-HL but was still significantly higher than in the surrounding sediment samples with 25 and 11 PLFAs, respectively. This seems to indicate that especially in the hyperarid desert locations, these rocks are more habitable and represent refuges for microbial life. Within the three microhabitats, the YS-HN sample showed the lowest amount and diversity of bacterial PLFAs. The reason for this is that the halite nodules were dominated, in addition to Cyanobacteria, by archaeal Halobacteria. Halobacterial membranes contain phospholipid ethers that are not recorded using the PLFA technique. Mono- and di-unsaturated FA (16:1Δ^9^, 18:1Δ^9^, 18:2Δ^9,12^) as well as methyl-branched FA (e.g., 10-Me-14:0) known from photolytic microorganisms such as Cyanobacteria [90] were detected in all three habitats; however, typical polyunsaturated FA were not found. The occurrence of 12-Me-16:0 confirmed the presence of *Rubrobacter* in the bacterial community of CS-HL and AL-GG [91].

### 3.7. Metabolites

We also analyzed the water-extractable metabolites via direct injection electrospray ionization Fourier-transform ion cyclotron resonance mass spectrometry (DI-ESI (−) FT-ICR-MS) to further test for microbial activity in the collected rock samples. The number of compounds recovered from the rock islands indicated fresh biological material decreasing from CS-HL to AL-GG and YS-HN involving mainly CHNO (amino acids, small peptides) and CHO type of molecules (fatty acids of increasing chain length and saturation; Figure 10b–d). However, the unconsolidated samples surrounding the rock islands contained a higher abundant signature of CHNOS elemental compositions, indicating long term geochemical processes with a typical geochemical footprint of natural organic humified matter superimposed by fresh organic material ([26]; Figure 10e,f).

## 4. Discussion

In this study, we employed a diverse array of tools to explore microbial life and specifically investigate putative active microbiota and processes across four distinct Atacama Desert locations.

Our results indicate that translucent rocks were inhabited by permanently active hypolithic microbial communities dominated by phototrophic Cyanobacteria and Chloroflexi. This was supported both by our extensive e- and iDNA results, which focused on iDNA recovered from intact cells, and metagenomic analyses, which showed actively replicating bacterial populations. Detected viruses and the putative host–virus interactions inferred from the sequence data also suggested activity and turnover of the microbial communities. This conclusion was further strongly supported by the results from the other methodologies used, including the ATP results, which showed that the ATP concentrations within cells (iATP) was about five orders of magnitudes higher in the hypoliths than in the gypsum crusts and halite nodules. Unfortunately, not enough biomass was recovered from the gypsum crusts and halite nodules for metagenomic analyses. However, the e- and iDNA results, together with the ATP, PLFA, and metabolomic analyses, still revealed a comprehensive picture on the microbial communities in these habitats, which led us to conclude that microbial communities within the gypsum crusts and halite nodules are only temporarily active (though sampling at different time periods would be needed to prove our conclusion). Further insights were gained pertaining to (1) aridity and habitat type driving microbial ecology and biomass turnover, (2) Cyanobacteria, particularly *Chroococcidiopsis* (Nostocales order), playing an important role as primary producers in the Atacama Desert ecosystem, and (3) the Atacama Desert harboring unique hotspots of microbial life.

### 4.1. Aridity and Habitat Type Drive Microbial Ecology and Biomass Turnover

The iDNA and PLFA based analyses revealed viable microbial communities with abundance and diversity changing across the three evaluated lithic habitats (Table 1, Figure 4 and Figure 6). We suggest that over time, the assemblages in each location were shaped by the environmental conditions, especially aridity, which led to an enrichment of mostly halophilic, UV-resistant, and phototrophic species. However, as supported by the microbial community composition in the iDNA and the limited presence of biomarkers (Figure 8 and Figure 9) in the more arid habitats, eventually only a minor, probably adapted part of these preselected organisms might have been able to maintain metabolic activity under the harsh environmental conditions, particularly in CS and AL hypoliths (CS-HL and AL-HL), whereas the majority of microorganisms in the halite noodles (YS-HN) and a substantial part of the gypsum crust (AL-GG) of the potentially viable microbial communities remained inactive or dormant (Figure 7). This kind of life strategy is to be expected, as it was found that in nutrient-poor ecosystems up to 40% of taxon richness is made up by dormant bacteria [92]. Dormancy is triggered by environmental cues and is most common in bacteria [93] but also occurring in eukaryotic and even multicellular organisms [94,95].

Hypolithic communities were characterized by a higher microbial biomass, were more diverse, and were dominated by Cyanobacteria of the genera *Aliterella* and *Chroococcidiopsaceae* (Table 1, Figure 6), known for their adaptability to extreme environmental conditions and endemic occurrence in hypoliths from hot and cold deserts [29,96,97,98]. Minor differences in diversity and composition between iDNA and eDNA pools indicate a higher biomass turnover [99,100], and together with the presence of biological molecules (ATP, metabolites, and PLFAs) and evidence for replication (Figure 7, Figure 8 and Figure 9) indicate hypolithic microbial communities to be active. This is a novel finding showing that the hypoliths indeed represent microbial hotspots of life in the Atacama Desert and likely in similar environmental settings as well.

Our findings were different for the more arid locations, AL and YS. Here we discovered a diverse array of microorganisms with distinct physiological characteristics with differences between eDNA and iDNA pools being more pronounced. AL communities were especially in the iDNA pool enriched in methylotrophic taxa, such as *Methylobacterium* and *Methylophilaceae*, suggesting methane and/or methanol represents two of the major carbon sources in this environment. Methanol is produced chemically in the atmosphere and in the oceans from where it is emitted into the air [101] and may then be carried into the Atacama Desert by wind/fog moving inland from the Pacific Ocean. Methylotrophic taxa could be identified both in the AL-GG eDNA and iDNA pools, indicating a higher turnover rate and by this a more active microbial community. We also observed photosynthetic activity in the AL samples, which may be attributed to the presence of, albeit believed to be non-phototrophic, members of the taxon Chloroflexi or may stem from algae that were present in the community, as suggested by Wierzchos et al. [32].

In the Yungay Salar halite nodule communities, both Cyanobacteria and halophilic archaea were present. Keystone species were *Halococcus* and bacteria such as *Salinibacter*, which were also found by other researchers [11,15,102]. Their abundance in the iDNA pool suggests they are able to survive in the harsh conditions, supporting earlier pulse-amplitude-modulation (PAM) fluorometry measurements by Davila et al. [103]. Both *Salinibacter* and members of the Halobacteriales have unique molecular adaptations allowing them to overcome the low water activity and the extremely high salt environment [104,105,106]. Microbial life may also benefit from the ability of halite nodules to undergo deliquescence by absorbing moisture from their surroundings and providing water to the halite microbial community in the form of concentrated brine [33,83,103,107]. However, the limited presence of these ASVs in the corresponding eDNA pool is an indicator for slow replication rates, pointing to a unique microbial community which can only be active from time to time, e.g., after a rare rain event or when fog drifts far inland. YS halite nodules represent an example of an endolithic habitat used as a last refugee by microbial life being pushed to the limit due to extreme environmental conditions. It may also be a suitable analog environment for the search for life on Mars.

### 4.2. Cyanobacteria, Particularly Chroococcidiopsis, Play an Important Role as Primary Producers in the Atacama Desert Ecosystem

Our results not only highlight the high abundance of photosynthetic Cyanobacteria in the analyzed hypoliths (Figure 4, Figure 5 and Figure 6) but also confirm their presence in the living part of the microbial communities at the more arid halite nodule site, indicating these microorganisms to be an important part of the local ecosystem. Thus, Cyanobacteria are not only a keystone species for the southern arid region of the Atacama Desert, as found by Lehnert et al. [27], but also for more northern areas and the hyperarid region of the Atacama. In general, Cyanobacteria have been identified as prominent inhabitants of soil gypsum in hot and cold deserts [15,32,34,108]. More specifically, comparable populations from the hyperarid Atacama Desert have been postulated to be photosynthetically active, as a chlorophyll signal could be measured [25]. These observations support our biomarker (ATP and PLFA) measurements and iRep data for hypolith populations (Figure 7, Figure 8 and Figure 9), implying that even at very salty and arid locations, Cyanobacteria populations are alive and metabolically active. Since genome replication (as indicated by iRep data) requires the acquisition of carbon, we conclude that the Cyanobacteria actively fix carbon in the Atacama Desert. This is further supported by the presence of a Rubisco form I enzyme encoded in all cyanobacterial draft genomes, indicating that they are all able to fixate carbon via the Calvin–Benson–Bassham (CBB) pathway [11,109].

Along with our 16S rRNA amplicon data, reconstruction and annotation of three *Chroococcidiopsis* (Nostocales order) metagenomic assembled genomes (MAGs) highlight the importance of this particular taxon in the hypolith habitats and provides additional, metagenomic insights into the metabolic potential of cyanobacterial communities in the Atacama Desert. The metagenomic investigations suggest that hypolithic *Chroococcidiopsis* populations have metabolic pathways for phototrophy and are resistant to desiccation, UV radiation, and high salt content, which may contribute to this species’ capability to thrive under the harsh conditions predominating in this microbial hotspot. Unlike in previously published *Chroococcidiopsis* genomes [11,83,84,110,111], no complete nitrogen fixation pathway could be identified in the recovered MAGs. Considering that nitrogen fixation is an energy intensive pathway and that some bins identified in the genome resolved metagenomics analyses contain parts of the pathway, it is possible that these hypolithic variants have adapted to a lifestyle resulting in the loss of some key *nif* genes. Sufficient nitrate and ammonium, as main sources of nitrogen for these communities, should be available due to accumulation of large nitrate deposits through atmospheric deposition over millions of years [112].

### 4.3. The Atacama Desert Harbors Unique Hotspots of Microbial Life

Based on our results, only the hypolith-colonized quartz rocks at the sampled CS and AL locations are continuously habitable sites, while the gypsum crusts and halite nodules become temporarily habitable after rare rain events or bouts of high humidity. The gypsum crusts and the salt nodules contain viable bacterial populations, which had recently produced more microbial metabolites (Figure 10), but less than those of the CS hypoliths. It appears that not much activity was occurring during our sampling event. Most of the species might only become active when a certain relative humidity threshold is reached. The surrounding desert sediments were highly influenced by geochemical degradation over the years and therefore contain more abundant natural organic humified matter (Figure 10) concurrent with the less diverse metagenome of the bacterial community (Figure 7). Microorganisms within the upper few cm of desert soil are subjected to extensive water stress and restricted access to organic substrates, resulting in long periods of stasis [113]. In contrast, the endolithic habitat provides protection from UV radiation and high solar irradiance, thermal buffering, protection from freeze–thaw events, physical stability, and enhanced water availability, ensuring sufficient protection for long-term survival in even harsh desert environments [17,114]. This does not only pertain to the rock habitats analyzed here, but also to other substrates colonized by endoliths, such as calcite, ignimbrites, granites, limestone, and quartz-rich shale [10,13,33]. Our study, which applied new molecular techniques, confirmed that the gypsum crusts and halite-dominated salt rocks surrounded by unconsolidated sediments can be considered islands of habitability or life as previously suggested by Warren-Rhodes et al. [24] and DiRuggiero et al. [21]. However, new insights gained show that they mostly represent transiently active life—not unlike that found in our previous work to be existing at a depth of 20–25 cm in the unconsolidated sediments of the Atacama Desert—microbial life, which was “awakened” by a major rainfall event [26].

Davila and Schulze-Makuch [115] proposed a predictable sequence of ecological transitions as a desert increases in aridity. While hypoliths are clearly a stand-out example of a thriving habitat, the gypsum crusts and the halite nodules are not. It appears that as hyperaridity approaches the physical limit observed on Earth, only the process of salt deliquescence provides enough water for microbes to thrive [103]. The near-surface environment containing hygroscopic salts can undergo deliquescence when the relative humidity (RH) exceeds the deliquescence relative humidity (DRH), which is 75% for halite [107]. When this occurs, microorganisms become active from their otherwise dormant state, which was not the case when we sampled the salt nodules. This process may also be relevant to the search for life on Mars given that Mars is even more arid than the driest deserts on Earth.

## 5. Conclusions

Our investigations, which used a novel DNA fractionation approach together with metagenomic analysis and other state-of-the-art methodologies, elucidated the living microbial populations and their interactions with the natural environment in three distinct desert habitats: hypolithic quartz rocks, gypsum crusts, and halite nodules. The microbial community inhabiting the hypolith sites was dominated by phototrophs, the gypsum crusts by methylotrophs and heterotrophic phototrophs, and the halite nodules by halo-tolerant and phototrophic communities, including archaea. While the colonized quartz rocks represent a habitat where microbial communities were active, this was only the case to a limited extent for the microbial communities of the gypsum crusts and not the case for the halite-dominated salt rocks. Due to the high numbers of microbial signals, however, we conclude that the latter two microhabitats are places of retreat for microbes, which can be reactivated when environmental conditions become more benign. Activity in the halite nodules especially appears to be based on a transient nature, with cycles of activity and dormancy. These three rock habitats inform us how microorganisms adapt to extreme aridity, as experienced in the Atacama Desert, and show that phototrophs play a key role in the ecology of this desert environment.

## Figures and Tables

**Figure 1 microorganisms-09-01038-f001:**
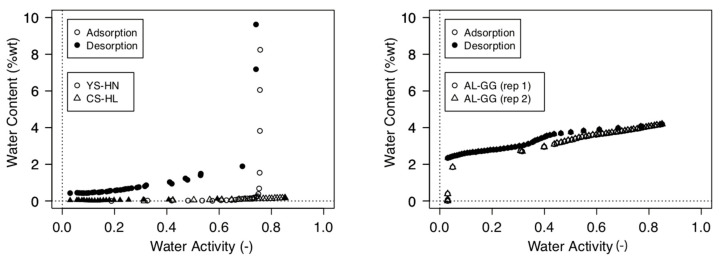
Moisture sorption isotherms of quartz rock (CS-HL), gypsum crusts (AL-GG), and halite nodules (YS-HN) at 25 °C. The halite nodules sample shows a steep increase in water content at a water activity of a_w_ = 0.753, corresponding to the water activity of a saturated NaCl solution at 25 °C.

**Figure 2 microorganisms-09-01038-f002:**
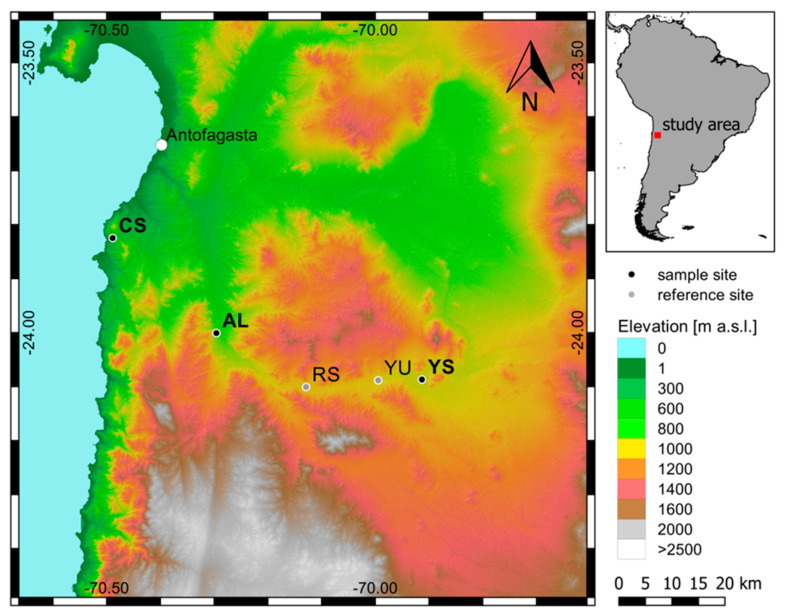
Map of sampling site locations. CS (Coastal Soil) is the least arid location, AL (Alluvial Fan) is arid to hyperarid, and the other locations YS (Yungay Salar), and the sediment background locations RS (Red Sands) and YU (Yungay) are within the hyperarid core of the Atacama Desert.

**Figure 3 microorganisms-09-01038-f003:**
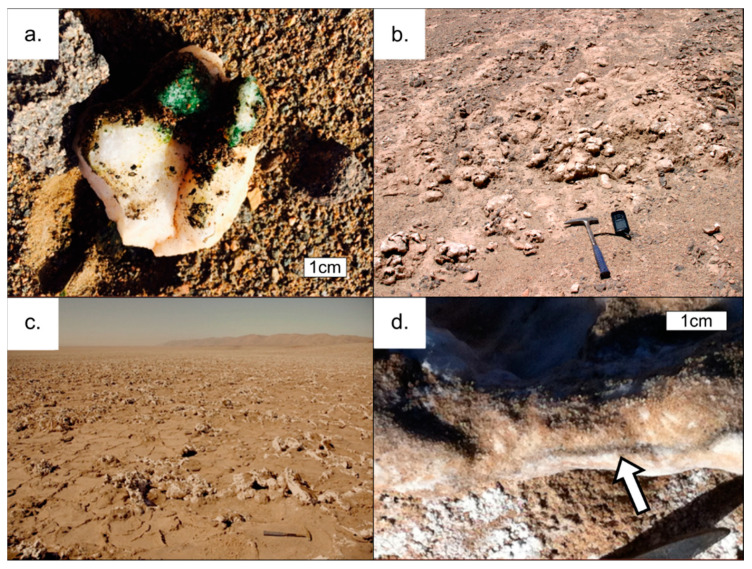
Rocks analyzed as microhabitats: (**a**) hypolithic quartz at the CS (Coastal Soil) location (this rock was turned upside down and had green Cyanobacteria on its underside), (**b**) gypsum crusts at the AL (Alluvial Fan) location, (**c**) YS (Yungay Salar) with (**d**) halite nodules magnified with visible scytonemin layer (with a thickness of 1–2 mm, see arrow) indicating microbial life. Quartz hypoliths (HL) were found at CS and AL, gypsum crusts (GG) at AL, and halite nodules (HN) only at YS (see Figure 2 for locations). Hammer for scale in (**b**,**c**) is 40 cm long.

**Figure 4 microorganisms-09-01038-f004:**
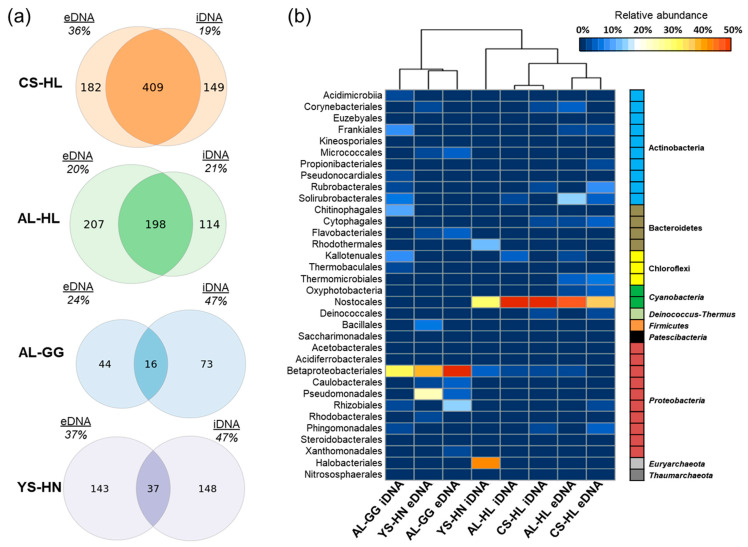
Venn diagrams showing the distribution of unique and shared ASVs across the eDNA and iDNA pools in each of the evaluated Atacama microhabitats as well as the percent of sequencing reads belonging to each fraction (**a**) and relative abundances of microorganisms (at order level > 1%) based on 16S rRNA sequencing in each of the analyzed samples (**b**) depicted values represent averages across 1–3 replicates. Color bar at right side indicates phylum for each of the shown orders. CS-HL are Costal Soil hypoliths, AL-HL are Alluvial Fan hypoliths, AL-GG are Alluvial Fan gypsum crusts, and YS-HN are Yungay Salar halite nodules.

**Figure 5 microorganisms-09-01038-f005:**
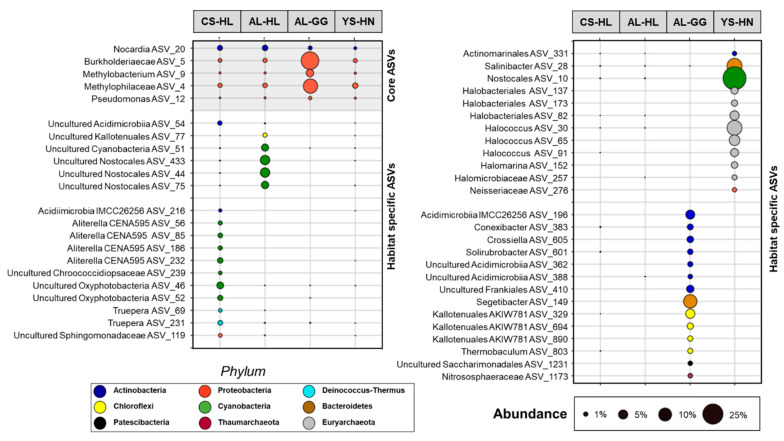
Relative abundance (as shown by bubble size) of “core ASVs” (gray shaded) and habitat specific ASVs (not shaded) across the iDNA pools of the four evaluated habitats. Core ASVs were detected in all four habitats, while specific ASVs were only identified in one of the four lithic habitats and are therefore considered unique to this environment. Colors indicate the corresponding phylum of each listed ASV. CS-HL are Costal Soil hypoliths, AL-HL are Alluvial Fan hypoliths, AL-GG are Alluvial Fan gypsum crusts, and YS-HN are Yungay Salar halite nodules.

**Figure 6 microorganisms-09-01038-f006:**
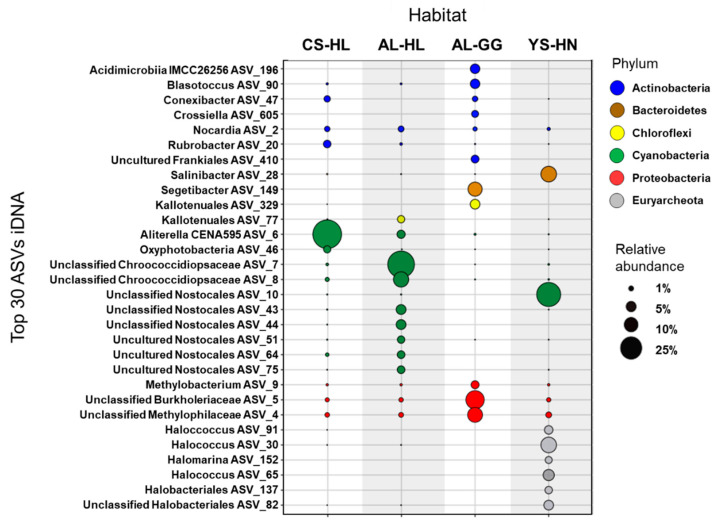
Top 30 ASVs across the iDNA pools of the four lithic microhabitats. CS-HL are Costal Soil hypoliths, AL-HL are Alluvial Fan hypoliths, AL-GG are Alluvial Fan gypsum crusts, and YS-HN are Yungay Salar halite nodules. Bubble size indicates average relative abundance of each ASV, while colors represent phylum assignment of the depicted ASVs.

**Figure 7 microorganisms-09-01038-f007:**
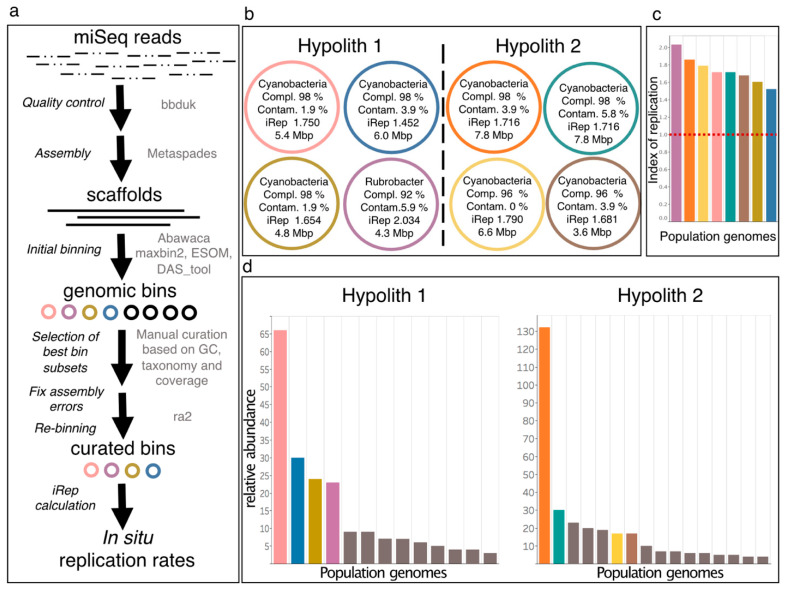
Genome-resolved metagenomic analysis of hypoliths from site CS (Coastal Soil). (**a**). Metagenomic analysis pipeline (right of the arrow lists the tools used). (**b**) Statistics of retrieved high-quality genomic bins. (**c**) In situ replication rates of retrieved genomic bins with colors corresponding to panel, red dashed line at iRep of 1, above which values indicate average replication rate for the populations with one origin of replication. (**d**) Community structure within hypoliths based on the relative abundance of the *rpS3* marker gene. Colors correspond to the genomes depicted in panel b with greyed out species having an associated draft genome without an iRep value. For further information, see also Appendix A.

**Figure 8 microorganisms-09-01038-f008:**
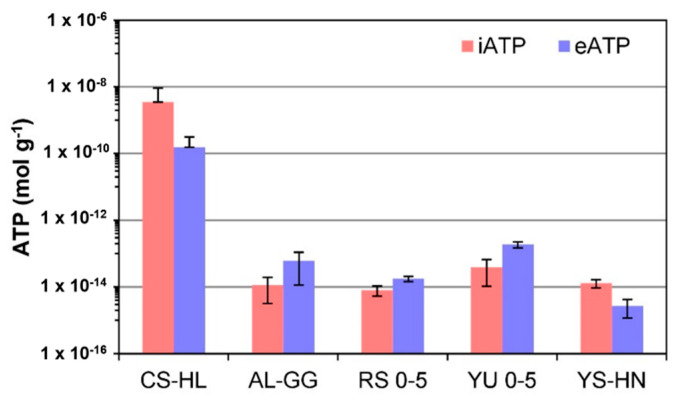
Concentrations of intracellular ATP (iATP) and extracellular ATP (eATP) (both *n* = 3). CS-HL (Coastal Soil hypoliths), AL-GG (Alluvial Fan gypsum crusts), and YS-HN (Yungay Salar halite nodules) represent “rock islands”, while RS (Red Sands) 0–5 and YU(Yungay) 0–5 denote surrounding sediments.

**Figure 9 microorganisms-09-01038-f009:**
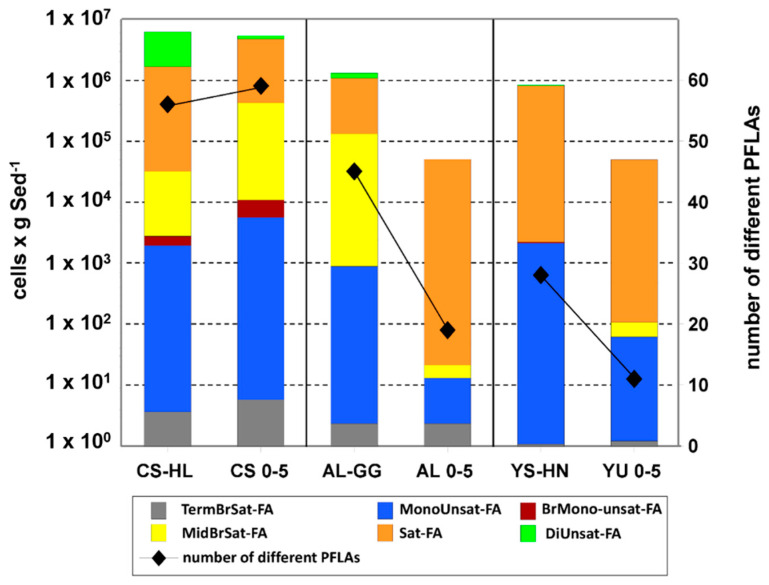
Bacterial cell biomass in cells × gSed^−1^ of the three microhabitats (CS-HL, AL-GG, and YS-HN) and their surrounding arid to hyperarid sediments (CS, AL, and YU 0–5 denoting 0–5 cm surface samples of the Coastal Soil, Alluvial Fan, and Yungay study sites) calculated from phospholipid fatty acid (PLFA) assessment [73]. The subdivisions of the bars represent percentage proportions of different structural PLFA types in the investigated samples. The black diamonds show the different numbers of individual PLFAs in the samples.

**Figure 10 microorganisms-09-01038-f010:**
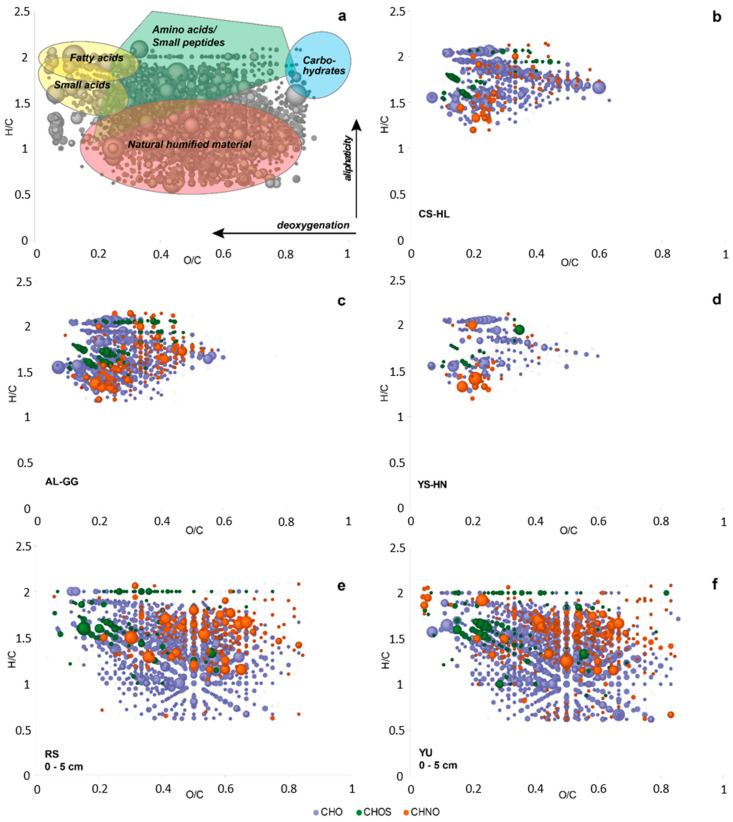
The positions of chemical classes (colored areas) are depicted in compositional space of a van Krevelen diagram according to their hydrogen to carbon (H/C) versus oxygen to carbon (O/C) ratios in (**a**). The elemental compositions are plotted as CHO (blue), CHOS (green), and CHNO (orange), and bubble sizes depict mass signal intensities. Highly aliphatic compounds are mostly presented in the upper area (H/C ratio *>* 1) and aromatic compounds in the lower (H/C ratio < 1). In this figure, we highlighted molecules of biological origin and the central zone of oxygenated aromatics corresponding to natural organic humified material. Van Krevelen diagrams of assigned molecular formulae of rock extracts from (**b**) Coastal Soil hypolith (CS-HL), (**c**) Alluvial Fan gypsum crusts (AL-GG), and (**d**) Yungay Salar halite nodules (YS-HN) have mostly a CHO elemental composition of biological origin compared to typical traces of more abundant CHNOS elemental compositions observed in sediments at the (**e**) Red Sands (RS) and (**f**) Yungay (YU) sites, showing more molecules of geochemical origin (typically humified material) within 0–5 cm depth.

**Table 1 microorganisms-09-01038-t001:** The 16S rRNA gene copy numbers per gram of material and Shannon diversity indices (rarefied to 69,772 sequence reads, see Methods) section across the e- and iDNA pools of the four sampling locations and the unconsolidated sediments collected around the lithic sites. HL = hypoliths, GG = gypsum crusts, HN = halite nodules. Sediment samples were taken at a depth of 0 to 5 cm. Errors represent standard deviations.

*Habitat*	16S rRNA Gene Copies		Shannon Index	
	iDNA	eDNA	iDNA	eDNA
*Coastal Soil-HL*	4.1 × 10^8^	2.3 × 10^9^	3.09	4.27
	±3.9 × 10^8^	±1.9 × 10^9^	±0.83	±0.86
*Alluvial Fan-HL*	1.8 × 10^9^	1.0 × 10^10^	2.89	3.51
	±2.2 × 10^8^	±9.6 × 10^8^	±0.53	±0.08
*Alluvial Fan-GG*	1.4 × 10^3^	2.1 × 10^3^	3.45 *	3.02 *
	±8.4 × 10^2^	±6.5 × 10^2^		
*Yungay Salar-HN*	3.8 × 10^4^	6.2 × 10^3^	2.43	2.91
	±2.9 × 10^4^	±2.3 × 10^3^	±0.39	±0.93
*Coastal Soil 0–5*	6.6 × 10^5^	3.9 × 10^7^	6.42	6.36
	±4.9 × 10^5^	±1.7 × 10^7^	±0.06	±0.05
*Alluvial Fan 0–5*	1.5 × 10^3^	NA	4.12	4.25
	±3.1 × 10^2^		±0.19	±0.52
*Red Sands 0–5*	2.8 × 10^3^	NA	3.84	3.72
	±1.2 × 10^3^		±0.27	±0.53
*Yungay Salar 0–5*	1.0 × 10^4^	NA	4.01	4.29
	±3.3 × 10^2^		±0.32	±0.13

* only one replicate; NA: not analyzed.

## Data Availability

Data is contained within the article and Appendix A. Sequencing reads from this study were deposited in the European Nucleotide Archive under the project accession number PRJEB43972.

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
