# Peer review of "Microbial Hotspots in Lithic Microhabitats Inferred from DNA Fractionation and Metagenomics in the Atacama Desert"

_microorganisms, 2021, doi:10.3390/microorganisms9051038_

Round 1

Reviewer 1 Report

This is an interesting study to explore active and dead microbiomes in lithic microhabitats of the extremely dry Atacama Desert. A lot of state-of-the-art technologies were applied, such as e-/i-DNA extraction, ATP analysis, and metabolite analysis. Introduction is well organized and informed. Methods section is comprehensive and professional. This study sheds light on the turnover rates of microbial materials and the tole of Cyanobacteria.

Could help if the continuous line numbers were added. I am happy with this manuscript. Just some trivial points for improvement:

Pg 5. Diversity and statistical analyses subsection. Past3 was mentioned before PAST 3.17 [49]. I guess they are the same software. If so, please be consistent; otherwise, please clarify. And “positive control” was mentioned in this subsection. Could you please also clarify what it is, since it may help understand why the cut-off number is based on it?

Pg 8. ESI(-) FT-ICR-MS analysis subsection. I am not sure if I can understand what (N rule; O/C ratio ≥ 1; H/C ratio ≤ 2n+2 (CnHn+2), double bond equivalents) are, especially the formula CnHn+2. If it is not mistyping, should be helpful to add slightly more information.

Pg 9. Should be better to mention the citation [26] in the beginning of the 3.1 section, otherwise it will confuse readers when the mineralogical analysis was performed as it is not in the Methods.

Pg 11. The full name of acronym ASV can be addressed when it appeared the first time, like in Pg 5.

Pg 11 and 21. PFLA should be PLFA.

Pg 15. “low-quality” is the only one with dash in between.

Pg 17 last line. 18:2Δ9,12 is not mono-unsaturated FA.

Pg 20. “order of magnitudes” should be “orders of magnitude”.

The last two paragraphs of the “Aridity and habitat type drive microbial ecology and biomass turnover” subsection of Discussion will be more helpful if the corresponding Figures or Tables could be referred to. Similarly, please cite related Figures and Tables in “Cyanobacteria, particularly Chroococcidiopsis, play an important role as primary producers in the Atacama Desert ecosystem”.

Table 1 caption. I think Experimental Procedures only appeared here, while Pg 13 referred to it as Methods. Also, why those without eDNA copy numbers have Shannon index?

Table S1 caption. Should “rimming” be “trimming”?

The font of Table S2 caption is different from others.

The format of subtitles in Discussion is not consistent with that in previous sections, easy to get misunderstood as an incomplete paragraph.

In Figure 4a, what are the percentages above Venn diagrams? They are not proportional to the numbers within circles.

Figures 7b and 7d are a little difficult to understand. Figure S3 has seven red-highlighted hypolith sample 1, why Figure 7 only illustrated four of them. And how the order of those greyed-out bars in Figure 7d determined? Why there are two types of grey colors (light and dark grey)? Could you please talk more about this analysis, maybe in Methods or wherever suitable?

Should the scale of the bubble size be specified in Figure 10 as a legend?

Can you check all Cyanobacteria to make sure the first letter is capitalized? Free feel to leave its adjective “cyanobacterial” all lower case. Similar things to “Archaea” and “archaeal”.

Author Response

Reviewer 1:

This is an interesting study to explore active and dead microbiomes in lithic microhabitats of the extremely dry Atacama Desert. A lot of state-of-the-art technologies were applied, such as e-/i-DNA extraction, ATP analysis, and metabolite analysis. Introduction is well organized and informed. Methods section is comprehensive and professional. This study sheds light on the turnover rates of microbial materials and the tole of Cyanobacteria.

Thank you

Could help if the continuous line numbers were added. I am happy with this manuscript. Just some trivial points for improvement:

Pg 5. Diversity and statistical analyses subsection. Past3 was mentioned before PAST 3.17 [49]. I guess they are the same software. If so, please be consistent; otherwise, please clarify. And “positive control” was mentioned in this subsection. Could you please also clarify what it is, since it may help understand why the cut-off number is based on it?

Thank you for your comment. We have changed the terminology from past3 to PAST 3.17 to be consistent. We sequenced and processed a separate positive control sample as an indicator for the efficacy and accuracy of the sequencing run and bioinformatic analysis.  We used E. coli DNA for this approach and updated our manuscript to include this information.

Pg 8. ESI(-) FT-ICR-MS analysis subsection. I am not sure if I can understand what (N rule; O/C ratio ≥ 1; H/C ratio ≤ 2n+2 (CnHn+2), double bond equivalents) are, especially the formula CnHn+2. If it is not mistyping, should be helpful to add slightly more information.

We have revised (N rule; O/C ratio ≥ 1; H/C ratio ≤ 2n+2 (CnHn+2), double bond equivalents)

to

(N rule; O/C ratio ≥ 1; H/C ratio ≤ 2n+2 ( maximum possible carbon saturation, with n defined as CnHn+2 for any formula), double bond equivalents)

to make it more clear.

Pg 9. Should be better to mention the citation [26] in the beginning of the 3.1 section, otherwise it will confuse readers when the mineralogical analysis was performed as it is not in the Methods.

We added a methods section on the mineralogy to describe how the analyses results were obtained. For the sediments we added citation [26]

Pg 11. The full name of acronym ASV can be addressed when it appeared the first time, like in Pg 5.

Thank you, we included the full name of ASV at its first mention in the manuscript.

Pg 11 and 21. PFLA should be PLFA.

Thank you for catching this. It is now corrected

Pg 15. “low-quality” is the only one with dash in between.

We changed this to match the other occurrences of this term.

Pg 17 last line. 18:2Δ9,12 is not mono-unsaturated FA.

Thank you, this has been corrected to "Mono- and di-unsaturated FA (.....) as well as methyl-branched FA.....

Pg 20. “order of magnitudes” should be “orders of magnitude”.

We appreciate your input, we changed the term to “orders of magnitude”.

The last two paragraphs of the “Aridity and habitat type drive microbial ecology and biomass turnover” subsection of Discussion will be more helpful if the corresponding Figures or Tables could be referred to. Similarly, please cite related Figures and Tables in “Cyanobacteria, particularly Chroococcidiopsis, play an important role as primary producers in the Atacama Desert ecosystem”.

Thank you for your comment. We have added referrals to the corresponding tables and figures to our manuscript.

Table 1 caption. I think Experimental Procedures only appeared here, while Pg 13 referred to it as Methods. Also, why those without eDNA copy numbers have Shannon index?

We have changed the term from “experimental procedures” to “methods”.

We were able to obtain 16S rRNA amplicon data for the eDNA pool from the more arid locations. Overall, these samples were low in biomass, but the extracted amount turned out to be sufficient for sequencing. Unfortunately, we were not able to get qPCR data from these samples. The qPCR conditions are slightly different than the conditions for amplicon PCR intended for sequencing. Potential reasons could be the low biomass or inhibition.

Table S1 caption. Should “rimming” be “trimming”?

This has been corrected

The font of Table S2 caption is different from others.

This has been changed

The format of subtitles in Discussion is not consistent with that in previous sections, easy to get misunderstood as an incomplete paragraph.

This has been fixed

In Figure 4a, what are the percentages above Venn diagrams? They are not proportional to the numbers within circles.

Thank you for your comment. The % in the Venn diagram describes the % of sequencing reads belonging to this fraction. We have added a description of this terminology to our manuscript.

Figures 7b and 7d are a little difficult to understand. Figure S3 has seven red-highlighted hypolith sample 1, why Figure 7 only illustrated four of them. And how the order of those greyed-out bars in Figure 7d determined? Why there are two types of grey colors (light and dark grey)? Could you please talk more about this analysis, maybe in Methods or wherever suitable?

Thank you for pointing these out. The illustrated genomes in Figure 7 correspond to high quality bins as iRep analysis was only performed for the high quality bins. During revision, we realized that this was not clear in the methods and we have edited the relevant section as follows: "Completeness and contamination of final bins were determined using CheckM [62] and bins were considered high-quality for completeness >90 % and contamination < 10%. In situ genome replication rates (iRep) were calculated for high quality bins allowing a maximum of three mismatches per mapped read." The order of the bars in figure 7d is determined by the rank abundance of rpS3 genes in decreasing order, and we have changed the shades of grey to unify them to a single shade."

Should the scale of the bubble size be specified in Figure 10 as a legend?

We feel that this would make the figures too complicated and usually the scale is not given in this more qualitative presentations, but if the reviewer insist we can do so

Can you check all Cyanobacteria to make sure the first letter is capitalized? Free feel to leave its adjective “cyanobacterial” all lower case. Similar things to “Archaea” and “archaeal”.

We have edited the manuscript and capitalized the term “Cyanobacteria” and “Archaea” where appropriate.

Reviewer 2 Report

Schulze-Makuch, et al. “Microbial hotspots in lithic microhabitats inferred from DNA fractionation and metagenomics in the Atacama Desert”

The authors employed a variety of techniques including differentiating between intracellular and extracellular DNA, as well as metagenomics to determine the active and non-active diversity of bacteria and archaea in lithic hotspots across the Atacama.

Comments to the authors:

Page 2: It might serve the authors to provide a more in-depth discussion of iDNA vs. eDNA for the reader, rather than having the reader look up another paper. This would provide the authors with additional explanations backing up their claims about active and past microbial communities. For instance, I am left wondering more about how eDNA implies past colonization – how far in the past? Could it imply a seasonal shift in microbial community?

Page 9, Figure 2: Why is Red Sands location included in the map when no samples were retrieved from that site?

Page 10, Table 1: Why is data from Red Sands location included in Table 1 when it is not discussed in the text?

Page 13: Are there biases associated with the iDNA/eDNA extraction methods/primers used that could be skewing the diversity of the microbial community?

Page 15: Provide citation for “…although published Chroococcidiopsis genomes (i.e. Chroococcidiopsis thermalis CP003597) found in other environments have nitrogen fixation pathways present and Chroococcidiopsis has been thought to be a nitrogen fixing cyanobacteria.”

Page 17, Section 3.5: “Based on ATP analyses only the hypoliths appear to be a continually active habitat.” Wouldn’t a time-series of multiple samples over various timepoints be required to come to this conclusion?

Page 17, Section 3.6: Multiple times, the word “special” is used to describe the rock samples, which seems out of place.

Figures 8, 10: Red Sands data are included in these figures but not discussed in the text.

Page 20: Following off the comment from page 13, are there biases in the extraction methods or bioinformatics pipeline that could be skewing the assessment of microbial diversity?

Page 20: “…which indicated actively replicating bacterial populations, carbon fixation for biomass production, and metabolic activity.” I don’t think that carbon fixation and active metabolic activity can specifically be mentioned here, when they are each implied by the discovery of actively replicating bacteria. Were any attempts made to look at specific genes within the assembled MAGs that code for carbon fixation? I don’t think that active carbon fixation can be inferred without also including metatranscriptomics analyses or at least looking for the presence of these genes in the assembled MAGs. Note: I see on page 21 some discussion of carbon/carbon fixation via the CBB pathway. Thus, I think the above mentioned conclusions should come after the discussion of the ability for carbon fixation in the community.

Page 20: “…which led us to conclude that microbial communities within the gypsum crusts and halite nodules are only temporarily active.” Again, I don’t know if this can be said without a time series.

Page 22: “Based on our results, only the hypolith-colonized…become temporarily habitable after rare rain events or bouts of high humidity.” Again, I don’t know if this can be concluded without further data.

Overall, the discussion hits on the main topics included in the manuscript and does a satisfactory job of addressing the conclusions of the paper.

Author Response

Reviewer 2:

Schulze-Makuch, et al. “Microbial hotspots in lithic microhabitats inferred from DNA fractionation and metagenomics in the Atacama Desert”

The authors employed a variety of techniques including differentiating between intracellular and extracellular DNA, as well as metagenomics to determine the active and non-active diversity of bacteria and archaea in lithic hotspots across the Atacama.

Comments to the authors:

Page 2: It might serve the authors to provide a more in-depth discussion of iDNA vs. eDNA for the reader, rather than having the reader look up another paper. This would provide the authors with additional explanations backing up their claims about active and past microbial communities. For instance, I am left wondering more about how eDNA implies past colonization – how far in the past? Could it imply a seasonal shift in microbial community?

We appreciate your input. We have added a more detailed description of our eDNA and iDNA method and also added a brief background on iDNA and eDNA. As the majority of the eDNA is believed to stem from dead, lysed cells, it is considered an indicator for past microbial communities and thus provides a glimpse at what type of microorganism were an active part of colonization in the past.  There are, however, also other sources of eDNA; such as active secretion or horizontal gene transfer across living cells, and therefore eDNA cannot exclusively be considered a representation of dead cells.  As the major goal of this study was to describe the active community (represented by the iDNA fraction) and the eDNA evaluation is considered a complimentary analysis we believe the applied method to be satisfactory for the here described study.  Finally, it is difficult to discern the exact age of the eDNA in the samples. Under the right circumstances DNA can be preserved for long periods of time, however DNA can also be degraded or utilized as an energy source by microorganisms. Altogether, this means eDNA could provide some information on seasonal variability, but can not be used as a reliable indicator.

Page 9, Figure 2: Why is Red Sands location included in the map when no samples were retrieved from that site?

A sediment sample from RS was obtained (as well as from CS, AL, And YU) for comparison purposes. The mineralogical and water content results of these samples are now also added at the end of Section 1. It is important to include the sediment samples in order to show that the lithic samples are microbial hotspots surrounded by unconsolidated sediment within the Atacama Desert

Page 10, Table 1: Why is data from Red Sands location included in Table 1 when it is not discussed in the text?

It is discussed in the text, now elaborated on in the e/iDNA section (at the end), and also in the Discussion section when we compare the lithic habitats to the surrounding unconsolidated sediments

Page 13: Are there biases associated with the iDNA/eDNA extraction methods/primers used that could be skewing the diversity of the microbial community?

Thank you for your comment. There is always bias associated with selecting a specific extraction method and/or a specific set of primers. Here, we applied a relatively new method (Alawi et al., 2014) to obtain separate iDNA and eDNA pools.  The method was tested for its reliability and has been used in previous studies. We have added a detailed description of the method to our manuscript.

The 16S rRNA primers we used (515F -806R) are commonly used and have been applied in numerous environmental microbiome studies around the world. Selecting these primers allows more direct comparison of our data to previous studies and allows other researchers to include the deposited data in cross-comparison analyses.

Page 15: Provide citation for “…although published Chroococcidiopsis genomes (i.e. Chroococcidiopsis thermalis CP003597) found in other environments have nitrogen fixation pathways present and Chroococcidiopsis has been thought to be a nitrogen fixing cyanobacteria.”

The reference Shimura et al (2015) has been added

Page 17, Section 3.5: “Based on ATP analyses only the hypoliths appear to be a continually active habitat.” Wouldn’t a time-series of multiple samples over various timepoints be required to come to this conclusion?

We deleted the word “continually” in the sentence to address the reviewer´s concern

Page 17, Section 3.6: Multiple times, the word “special” is used to describe the rock samples, which seems out of place.

Agreed, the wording has been changed accordingly

Figures 8, 10: Red Sands data are included in these figures but not discussed in the text.

Actually, they are, but they are not referred as Red Sands, but as surrounding sediment samples to which Red Sands belongs (in case of Fig. 8 this is Red Sands (RS) and Yungay (YU). We think it is important to draw the comparison between the lithic samples and the unconsolidated ones, which is also further discussed in the Discussion Section

Page 20: Following off the comment from page 13, are there biases in the extraction methods or bioinformatics pipeline that could be skewing the assessment of microbial diversity?

Similar to the extraction methods, the pipeline we employed using Dada2 and Qime2 is commonly used in microbial genomic projects using 16S amplicon data. The biggest biases in the methodology could stem from the quality filtering parameters (q-score) and databases choice and alignment. Here, we followed the generally accepted guidelines and recommendations. We used the most comprehensive 16S rRNA database (SILVA v132) available at the time of processing.

Page 20: “…which indicated actively replicating bacterial populations, carbon fixation for biomass production, and metabolic activity.” I don’t think that carbon fixation and active metabolic activity can specifically be mentioned here, when they are each implied by the discovery of actively replicating bacteria. Were any attempts made to look at specific genes within the assembled MAGs that code for carbon fixation? I don’t think that active carbon fixation can be inferred without also including metatranscriptomics analyses or at least looking for the presence of these genes in the assembled MAGs. Note: I see on page 21 some discussion of carbon/carbon fixation via the CBB pathway. Thus, I think the above mentioned conclusions should come after the discussion of the ability for carbon fixation in the community.

We thank the reviewer for spotting this inconsistency. We have removed the phrase “carbon fixation for biomass production and metabolic activity” from this line.

Page 20: “…which led us to conclude that microbial communities within the gypsum crusts and halite nodules are only temporarily active.” Again, I don’t know if this can be said without a time series.

We think that this is still a fair statement given our accumulated evidence, but we agree that a time series would be preferable and need for a proof. Thus, we added in parenthesis “(though sampling at different time periods would be needed to proof our conclusion)”

Page 22: “Based on our results, only the hypolith-colonized…become temporarily habitable after rare rain events or bouts of high humidity.” Again, I don’t know if this can be concluded without further data.

We think that statement is justified because if the water activity drops below a certain limit the microbes within these habitats cannot metabolize anymore

Overall, the discussion hits on the main topics included in the manuscript and does a satisfactory job of addressing the conclusions of the paper.

Thank you